# 4-Methyl-2,4-bis(4-hydroxyphenyl)pent-1-ene, a Major Active Metabolite of Bisphenol A, Triggers Pancreatic β-Cell Death via a JNK/AMPKα Activation-Regulated Endoplasmic Reticulum Stress-Mediated Apoptotic Pathway

**DOI:** 10.3390/ijms22094379

**Published:** 2021-04-22

**Authors:** Cheng-Chin Huang, Ching-Yao Yang, Chin-Chuan Su, Kai-Min Fang, Cheng-Chieh Yen, Ching-Ting Lin, Jui-Min Liu, Kuan-I Lee, Ya-Wen Chen, Shing-Hwa Liu, Chun-Fa Huang

**Affiliations:** 1Department of Emergency, Taichung Tzu Chi Hospital, Buddhist Tzu Chi Medical Foundation, Taichung 427, Taiwan; cmcherkimo@yahoo.com.tw (C.-C.H.); andylau403@tzuchi.com.tw (J.-M.L.); leeguanto@gmail.com (K.-IL.); 2Department of Surgery, National Taiwan University Hospital, Taipei 100, Taiwan; cyang@ntuh.gov.tw; 3Department of Otorhinolaryngology, Head and Neck Surgery, Changhua Christian Hospital, Changhua County 500, Taiwan; 91334@cch.org.tw; 4School of Medicine, Kaohsiung Medical University, Kaohsiung 807, Taiwan; 5Department of Otolaryngology, Far Eastern Memorial Hospital, New Taipei City 220, Taiwan; u701048@gmail.com; 6Department of Occupational Safety and Health, College of Health Care and Management, Chung Shan Medical University, Taichung 402, Taiwan; ycj@csmu.edu.tw; 7School of Chinese Medicine, College of Chinese Medicine, China Medical University, Taichung 404, Taiwan; gingting@mail.cmu.edu.tw; 8Department of Physiology and Graduate Institute of Basic Medical Science, School of Medicine, College of Medicine, China Medical University, Taichung 404, Taiwan; ywc@mail.cmu.edu.tw; 9Institute of Toxicology, College of Medicine, National Taiwan University, Taipei 100, Taiwan; 10Department of Nursing, College of Medical and Health Science, Asia University, Taichung 413, Taiwan

**Keywords:** 4-methyl-2,4-bis(4-hydroxyphenyl)pent-1-ene (MBP), β-cells, apoptosis, ER stress, JNK, AMPKα

## Abstract

4-methyl-2,4-bis(4-hydroxyphenyl)pent-1-ene (MBP), a major active metabolite of bisphenol A (BPA), is generated in the mammalian liver. Some studies have suggested that MBP exerts greater toxicity than BPA. However, the mechanism underlying MBP-induced pancreatic β-cell cytotoxicity remains largely unclear. This study demonstrated the cytotoxicity of MBP in pancreatic β-cells and elucidated the cellular mechanism involved in MBP-induced β-cell death. Our results showed that MBP exposure significantly reduced cell viability, caused insulin secretion dysfunction, and induced apoptotic events including increased caspase-3 activity and the expression of active forms of caspase-3/-7/-9 and PARP protein. In addition, MBP triggered endoplasmic reticulum (ER) stress, as indicated by the upregulation of GRP 78, CHOP, and cleaved caspase-12 proteins. Pretreatment with 4-phenylbutyric acid (4-PBA; a pharmacological inhibitor of ER stress) markedly reversed MBP-induced ER stress and apoptosis-related signals. Furthermore, exposure to MBP significantly induced the protein phosphorylation of JNK and AMP-activated protein kinase (AMPK)α. Pretreatment of β-cells with pharmacological inhibitors for JNK (SP600125) and AMPK (compound C), respectively, effectively abrogated the MBP-induced apoptosis-related signals. Both JNK and AMPK inhibitors also suppressed the MBP-induced activation of JNK and AMPKα and of each other. In conclusion, these findings suggest that MBP exposure exerts cytotoxicity on β-cells via the interdependent activation of JNK and AMPKα, which regulates the downstream apoptotic signaling pathway.

## 1. Introduction

Diabetes mellitus (DM) is among the most prevalent chronic metabolic diseases, characterized by abnormally elevated levels of blood glucose and leading to serious injury to important organs [1]. It was estimated that 463 million people worldwide were living with diabetes in 2019 (9.3% of the global adult population). Projections indicate that the number of people with diabetes will increase to approximately 629 million globally by 2045 [2]. Epidemiological studies have reported that either genetic or environmental factors contribute to the risk of DM development. Particularly, among the environmental factors, growing evidence suggests that exposure to environmental endocrine-disrupting chemicals (EDCs) plays an important role in the pathophysiological processes of DM [3].

Bisphenol A (BPA) is a well-known and widely applied chemical that is used in the production of polycarbonate plastic bottles and food containers, and epoxy resins in the lining of metal cans [4,5]. The major source of human BPA exposure is the dietary intake of BPA that leak into foods and beverages. In an epidemiological study, BPA was detected in over 90% of all analyzed human urine samples, indicating widespread human exposure to BPA [6]. Moreover, BPA, as one of the EDCs, has been reported to induce pancreatic β-cell dysfunction and apoptosis, which is associated with the development of DM [7,8,9]. More importantly, 4-methyl-2,4-bis(4-hydroxyphenyl)pent-1-ene (MBP), an active metabolite of BPA, might accumulate in mammals upon BPA exposure, ingestion, and metabolism [6,10,11]. It has been reported that the toxic effects of MBP on medaka (*Oryzias latipes*) were estimated to be approximately 250-fold higher than those of BPA [12]. Okuda et al. [13] and Moreman et al. [14] reported that MBP exhibits stronger toxicological effects than BPA in in vivo experiments in zebrafish (*Danio rerio*) and ovariectomized (OVX) rats, in which MBP possessed a potency in estrogenic activity approximately 500–1000-fold more than BPA.

Recently, MBP exposure has been indicated to disrupt the development of both male and female reproductive organs in an avian embryo model, including retention of the Mullerian ducts and feminization of the left testicle in males and a diminished left ovary in females [15]. Furthermore, studies by Hirao-Suzuki et al. [16] and Liu et al. [17] reported that MBP exposure could induce pulmonary alveolar epithelial cell apoptosis and stimulate abnormal growth in breast cancer cells. A few studies have indicated that exposure to BPA could trigger pancreatic β-cell dysfunction and cytotoxicity, although MBP, an active metabolite of BPA, has not been investigated [18,19]. The effects of BPA metabolite MBP on β-cell function and growth still remain to be clarified. Therefore, we aimed to investigate the cytotoxic effects of MBP on pancreatic β-cell function and growth and their possible molecular mechanisms.

## 2. Results

### 2.1. BPA-Induced Cytotoxicity and Apoptosis in Pancreatic β-Cells

BPA-induced cytotoxic and apoptotic effects in β-cells were observed. As shown in Figure 1A, the treatment of pancreatic β-cell-derived RIN-m5F cells with BPA (25–300 μM) for 24 h substantially reduced cell viability, as determined by the MTT assay, in a concentration-dependent manner, with the effective concentrations ranging from 25 to 300 μM. By examining cytotoxicity using the LDH (an indicator of cytotoxicity) release assay, it was confirmed that BPA increased the cytotoxicity in β-cells in a concentration-dependent fashion (Figure 1B). In addition, BPA significantly induced the elevation of caspase-3 activity (Figure 1C) and increased the levels of cleaved caspase-3 and caspase-7 protein expression (Figure 1D). These results indicate that BPA can induce cytotoxicity through apoptosis in β-cells.

### 2.2. MBP-Induced Cell Apoptosis in RIN-m5F Cells

To investigate the potential damage of BPA metabolite MBP in β-cells, we examined the cytotoxic and apoptotic effects of MBP in RIN-m5F cells. As shown in Figure 2A, the treatment of RIN-m5F cells with MBP for 24 h markedly decreased cell viability in a concentration-dependent manner, with the effective concentrations ranging from 3 to 15 μM: 3 μM, 88.0% ± 8.4% of control; 5 μM, 55.5% ± 9.5% of control; 7 μM, 40.5% ± 4.5% of control; 10 μM, 23.4% ± 10.6% of control; and 15 μM, 12.7% ± 3.0% of control. Similar to the results of the cell viability assay, a concentration-dependent significant increase in LDH release was observed after MBP treatment for 24 h (3 μM, 22.5% ± 3.8% of total; 5 μM, 43.8% ± 5.8% of total; 7 μM, 57.4% ± 5.5% of total; 10 μM, 74.7% ± 6.9% of total; and 15 μM, 87.2% ± 5.3% of total; Figure 2B). Moreover, the caspase-3 activity dramatically increased after treatment of cells with MBP (3–7 μM) for 24 h. On the basis of these findings, the half maximal inhibitory concentration (IC_50_) of MBP in RIN-m5F cells was determined to be approximately 5 μM, which was used in subsequent experiments.

The cleavages of caspase-3, -7, and -9 and PARP proteins have been shown to be involved in chemical-induced cell apoptosis in various cell types. We next tested the effects of MBP on these apoptosis-related molecules in β-cells. As shown in Figure 2D, Western blot analysis showed an increase in the expression levels of cleaved forms of caspase-3, -7, and -9 and PARP proteins after 8 h of treatment of RIN-m5F cells with MBP (5 μM) and significantly continued to increase over 24 h (in a time-dependent manner). Furthermore, the exposure of RIN-m5F cells to 5 μM MBP for various time intervals (2–24 h) also resulted in a significant reduction in the cell viability at 8 h and a gradual dramatic reduction over 24 h (2 h, 101.6% ± 3.2% of control; 4 h, 100.9% ± 3.9% of control; 8 h, 83.6% ± 4.5% of control; 16 h, 69.6% ± 4.3% of control; and 24 h, 51.6% ± 6.2% of control) (Figure 2E).

The short-term effect of MBP on insulin secretion from β-cells was evaluated. As shown in Figure 2E, MBP at 3 and 5 μM effectively and significantly inhibited glucose-stimulated insulin secretion in RIN-m5F cells (1 μM, 98.4% ± 3.1% of control; 3 μM, 77.5% ± 6.0% of control; and 5μM, 61.1% ± 3.9% of control) after 4 h of exposure. These concentrations of MBP did not reduce the cell viability (99.8% ± 4.4%, 99.7% ± 2.6%, and 99.1% ± 3.5% of control for 1, 3, and 5 μM, respectively) (Figure 2F). These results indicate that MBP is capable of inducing cytotoxicity, apoptosis, and insulin secretion dysfunction in pancreatic β-cells and exhibits a greater toxicity than BPA.

### 2.3. MBP-Induced ER Stress Response in RIN-m5F Cells

ER stress has been reported to be involved in chemical-induced apoptosis in mammalian cells, including β-cells [20]. Thus, we next examined the effects of MBP on the expression of various ER stress markers in RIN-m5F cells. As shown in Figure 3A, the exposure of RIN-m5F cells to MBP (5 μM) markedly triggered the protein expression of ER stress-related molecules, including GRP 78 (but not that of GRP 94) and CHOP, as well as the degradation of full-length (55 kDa) caspase-12 (a downstream ER stress molecule) in a time-dependent manner (a statistical increase after 8 h exposure, and an obviously continued increase up to 24 h), as in the case of apoptosis-related molecules. To further confirm the relationship between MBP-induced β-cell apoptosis and the activation of ER stress, RIN-m5F cells were pretreated with a pharmacological inhibitor of ER stress (4-phenylbutyric acid, 4-PBA; 3 mM) for 1 h prior to MBP exposure and subsequently exposed to MBP (5 μM) for 24 h. The results showed that MBP-induced activation of ER stress-related molecules, including the upregulation of GRP 78, CHOP, and cleavage caspase-3 and 7 protein expression and degradation of pro-caspase-12, was effectively and significantly prevented by 4-PBA (Figure 3B). Furthermore, MBP-induced elevation of caspase-3 activity and reduction of cell viability were also significantly attenuated after pretreatment with 4-PBA (Figure 3C,D). These results imply that MBP can induce ER stress-regulated apoptosis, leading to β-cell death.

### 2.4. JNK and AMPK Signaling Played Crucial Roles in MBP-Induced β-Cell Apoptosis

JNK and AMPK-mediated signaling pathways play important roles in toxic chemical-induced apoptosis [20,21]. We next examined whether JNK/AMPK activation was involved in MBP- induced β-cell apoptosis. As shown in Figure 4, the levels of protein phosphorylation of JNK and AMPKα were significantly increased after exposure of RIN-m5F cells to MBP for 0.5–2 h. Pretreatment of cells with JNK inhibitor (SP600125, 10μM) for 1 h prior to MBP exposure significantly inhibited the levels of protein expression of both phosphorylated JNK and phosphorylated AMPKα (Figure 4B,*a*). Similarly, pretreatment with a specific AMPK inhibitor (compound C, 10 μM) markedly attenuated the activation of both AMPKα and JNK following a 2-h treatment with MBP (Figure 4B,*b*).

Furthermore, the activation levels of caspase-3 and -7, GRP 78, and CHOP protein induced following 24 h treatment with MBP were effectively reversed by pretreatment with a specific JNK inhibitor (SP600125, 10 μM) and AMPK inhibitor (compound C, 10 μM) (Figure 5A), along with the inhibition of cell viability (Figure 5B). These results suggest that the activation of both JNK and AMPK signaling-regulated apoptosis pathways participates in MBP-induced pancreatic β-cell death.

## 3. Discussion

Pancreatic β-cell insufficiency and apoptosis are the causes of DM [22,23]. Growing evidence has indicated that environmental pollutants are an important risk factor contributing to the DM epidemic [24,25]. Cadmium exposure could cause pancreatic β-cell apoptosis and death via oxidative stress-mediated JNK activation and Ca^2+^-triggered JNK/CHOP signaling pathways [21,26]. Exposure to low-dose tributyltin induced oxidative stress-triggered JNK-related pancreatic β-cell apoptosis in vitro and reversible hypoinsulinemic hyperglycemia in vivo, dysregulating β-cell function even under noncytotoxic doses [27,28]. BPA is one of the highest-production-volume chemicals in the world and is an environmental risk factor for the development of DM; humans are continually exposed to it via contaminated environments, plastic, and other products [29,30]. BPA has been detected at the concentrations of 21–17,200 μg/L (approximately 0.1–75.3 μM) in polluted aquatic environments such as river water and landfill leachate [31]. In resin-based composites and sealants in dentistry, BPA levels were reported to range from 0.5 to 84.4 μg per 100 mg of commercial product, and the saliva samples obtained after treatment using these products contained 3.3 to 30.0 μg/mL (approximately 14.5 to 131.4 μM) [32]. In some animal studies, long-term exposure to 0.1–10 mg/kg/day BPA decreased plasma insulin levels and increased the number of active caspase-3- positive cells in the pancreatic islets (approximately 0.44–43.8 μM/day), which accelerated DM development [33,34]. More importantly, MBP, as an active metabolite of BPA, is formed upon coincubation of BPA and liver microsomal and cytosolic fractions (S9 fraction from mammalian liver samples, including human samples) [11]. Consequently, it has been considered that BPA released into the aquatic environment, which gains access to the body through various routes, could be converted to MBP in the mammalian liver, triggering a considerably stronger toxicological effect than its parent compound BPA [6,35,36]. It has been shown through luciferase reporter assays in vitro that MBP is a more potent binder of the estrogen receptor than BPA [11,13]. Hirao-Suzuki and colleagues have also reported that repeated exposure to MBP, but not to BPA, aggressively stimulated abnormal proliferation in breast cancer cells through the activation of estrogen receptor β-dependent signaling [16]. More importantly, Ishibashi et al. [12] have shown that the 96-h median lethal concentrations of MBP and BPA in medaka (*Oryzias latipes*) were estimated to be 1640 and 13,900 μg/L (approximately 6.1 and 60.9 μM), respectively. A study by Okuda et al. [13] in an ovariectomized (OVX) female rat model indicated that MBP (1000 μg/kg/day (approximately 3.7 μM/day)) completely reversed the changes caused by OVX, as equivalent to the activity of 17β-estradiol 0.5 μg/kg/day, suggesting that MBP exhibited at least 500-fold higher estrogenic activity than BPA. Furthermore, exposure to MBP at 25 μg/L (approximately 0.1 μM) has been found to impair cardiovascular function and induce the development of vascular-cardiovascular disease states in zebrafish [36]. Huang et al. [37] and Liu et al. [17] have also demonstrated that MBP exposure (5–15 μM) induced dysfunction and apoptosis in pulmonary alveolar epithelial cells and neuronal cells. However, the cytotoxic effect and mechanism of action of MBP on β-cells remain unclear, especially with respect to its possible exposure concentration in mammalians. This study demonstrated for the first time that the treatment of RIN-m5F cells with MBP significantly induced cytotoxicity in a concentration-dependent manner (ranging from 3 to 15 μM; IC_50_ was approximately 5 μM), which was accompanied with insulin secretion dysfunction and apoptotic events. Moreover, the signaling mechanisms of both JNK and AMPK activation, which contribute to triggering ER stress-mediated apoptosis, were involved in MBP-induced β-cell death.

The ER is an important and essential cell organelle required for cell survival [38]. ER is also the major site for the synthesis, correct folding, and post-translational maturation of almost all membrane proteins [39]. The accumulation of unfolded, misfolded, or aggregated proteins in the ER lumen, called ER stress, activates the unfolded protein response (UPR) to resolve the protein-folding defect [40,41]. However, during excessive and long-term upregulation of UPR or severe ER stress, the protective mechanisms activated by UPR are insufficient to restore normal ER function/homeostasis, leading to cell damage and death by apoptosis [42,43]. ER stress induces the expression of GRP 78 and GRP 94, the major ER-localized chaperones and the most abundant glycoproteins in the ER, and the activation of GRPs regulates toxic insult-induced cell apoptosis [44,45]. Moreover, the activation of CHOP (also known as GADD153) plays an important role in ER stress-induced mammalian cell apoptosis [46,47]. CHOP belongs to the CCAAT/enhancer-binding protein (C/EBP) family of transfection factors, which has been implicated in the regulation of processes associated with cellular proliferation, differentiation, and energy metabolism [48]. ER stress is associated with numerous pathophysiological conditions in various human diseases, including DM [49,50]. Growing evidence has shown that toxic chemical-induced ER stress is involved in the process of apoptosis in β-cells, skeletal muscle-derived myoblasts, and pulmonary alveolar epithelial cells [17,20,21,51,52,53]. Recently, Liu et al. [17] and Huang et al. [37] indicated that MBP exposure significantly induced the AMPK/ERK/Akt signal-regulated ER stress-triggered apoptotic pathway, resulting in type 2 pulmonary alveolar epithelial cell and neuronal cell death; however, this effect has not been evaluated in β-cells. The results of this study revealed that the treatment of RIN-m5F cells with MBP markedly increased the activation of ER stress-related molecules, including GRP 78, CHOP, and caspase-12. Pretreatment of β-cells with an ER stress inhibitor 4-PBA effectively prevented the MBP-induced protein expression of GRP 78, CHOP, and caspase-12, as well as apoptotic events (including the expression of the cleaved forms of caspase-3 and caspase-7 and the increase in caspase-3 activity). These results indicate that downstream ER stress activation-regulated apoptosis plays a crucial role in MBP-induced β-cell death.

JNKs belong to the superfamily of MAP kinases, which have been shown to play a crucial role in the regulation of cell proliferation, differentiation, and apoptosis [54]. Activation of the JNK pathway induced by diverse stimuli (such as cytokines, Aβ peptides, oxidative stress, or toxic chemicals) has been identified as a critical factor in pathological cell death, associated with the development of many diseases, including DM [55,56]. Accumulating evidence suggests that a marked activation of the JNK signal in pancreatic β-cells exposed to toxic insults subsequently leads to β-cell dysfunction and apoptotic cell death [20,21,28], implying that JNK plays an important role in β-cell damage. Furthermore, AMPK is a multimeric serine/threonine protein kinase, composed of α- (catalytic), β- (scaffold), and γ- (regulatory) subunits [57]. It has been confirmed that AMPK, a master energy sensor and a coordinator of cell growth/proliferation, is also a key regulator of cell apoptosis under pathological conditions [58,59]. Increasing evidence demonstrates the pivotal role of AMPK in enhancing pancreatic β-cell death and apoptosis via the regulation of various signaling molecules, including JNK, which is induced by toxic chemicals or the pathophysiological processes of DM [20,60,61]. For example, it has been shown that stimulation of AMPK activation by the adenosine analogue AICA-riboside (AICAR) or metformin could induce the apoptosis of insulin-producing β-cells, which involved JNK signaling [62,63]. Yang et al. [20] reported that molybdenum exposure could cause pancreatic β-cell dysfunction and death via JNK and AMPK activation-interdependent downstream-regulated apoptosis pathways. Lin et al. [19] have suggested that BPA-triggered β-cell apoptosis may be mediated via the mitochondrial pathway. However, to our knowledge, no study has elucidated the role of JNK and AMPK activation in MBP-induced β-cell apoptosis. In our study, we found that treatment of RIN-m5F cells with MBP significantly induced the phosphorylation of both JNK1/2 and AMPKα proteins. Pretreatment of β-cells with JNK inhibitor (SP600125) effectively prevented MBP-induced apoptotic- and ER stress-related responses (including the suppression of caspase-3, caspase-7, GRP 78, and CHOP activation), and the phosphorylation of the JNK and AMPKα proteins. Similarly, pretreatment of β-cells with an AMPK inhibitor (compound C) significantly abrogated MBP-induced β-cell apoptosis and ER stress responses, concomitant with the marked inhibition of AMPKα and JNK activation. These results indicate that both JNK- and AMPK-mediated signals are interdependent and play critical roles in the downstream regulation of ER stress, triggering an apoptosis pathway implicated in MBP-treated pancreatic β-cell death.

## 4. Materials and Methods

### 4.1. Materials

Unless otherwise specified, all chemicals (including MBP) and laboratory plastic wares were purchased from Sigma-Aldrich (St. Louis, MO, USA) and Falcon Labware (Becton, Dickinson and Company, Franklin Lakes, NJ, USA), respectively. RPMI 1640 medium, fetal bovine serum (FBS), and antibiotics were purchased from Gibco/Invitrogen (Thermo Fisher Scientific Inc., Waltham, MA, USA). Mouse- or rabbit- monoclonal antibodies specific for caspase-3, caspase-7, caspase-9, PARP, phosphorylated (*p*)-AMPKα, GRP 78, GRP 94, AMPKα, CHOP, and β-actin, and secondary antibodies (anti-mouse or anti-rabbit IgG-conjugated to horseradish peroxidase (HRP)) were purchased from Cell Signaling Technology (Cell Signaling Technology, Danvers, MA, USA); caspase-12 was purchased from Santa Cruz Biotechnology (Santa Cruz Biotechnology, Santa Cruz, CA, USA); and antibodies specific for *p*-JNK and JNK-1 were purchased from Abclonal (Woburn, MA, USA).

### 4.2. Pancreatic β-Cell-Derived RIN-m5F Cell Culture

RIN-m5F cells, a rat pancreatic islet β-cell line, were capable of producing and secreting insulin. RIN-m5F cells were purchased from the American Type Culture Collection (ATCC, CRL-11605) and cultured in a humidified chamber with a 5% CO_2_–95% air mixture at 37 °C and maintained in RPMI 1640 medium supplemented with 10% fetal bovine serum (FBS) and antibiotics (100 U/mL of penicillin and 100 μg/mL of streptomycin).

### 4.3. Cell Viability and Cytotoxicity Assay

The cells were washed with fresh medium and cultured in 96-well plates (2 × 10^4^ cells/well) and then stimulated with BPA (25–300 μM) or MBP (1–15 μM) for 24 h. After incubation, the medium was aspirated and the cells were incubated with fresh medium containing 0.2 mg/mL 3-(4,5-dimethyl thiazol-2-yl-)-2,5-diphenyl tetrazolium bromide (MTT). After 4 h, the medium was removed and the blue formazan crystals were dissolved in 100 μL of dimethyl sulfoxide (DMSO). Absorbance at 570 nm was measured using a Bio-Tek uQuant Microplate Reader (MTX Lab Systems, Winooski, VT, USA).

Cytotoxicity was determined on the basis of the amount of lactate dehydrogenase (LDH) that leaked out of the cytosol of damaged cells into the medium after BPA or MBP exposure for 24 h. The cells were seeded as described for the MTT assay. After 24 h of treatment, 40 μL of the supernatant was added to a new 96-well plate to determine the LDH release, and a cell lysis buffer was also added to the positive control group to determine the total LDH. The amount of LDH from the cells was quantified using the LDH Cytotoxicity Assay Kit (BioVision, Inc., Milpitas, CA, USA) according to the manufacturer’s instructions. Absorbance at 450 nm was measured using a Bio-Tek uQuant Microplate Reader (MTX Lab Systems, Winooski, VT, USA).

### 4.4. Determination of Insulin Secretion

To measure the amount of insulin secretion in RIN-m5F cells after exposure to MBP, the cells were incubated in Krebs–Ringer buffer (KRB) with 2.8 or 20 mM glucose to stimulate insulin secretion, and the supernatant of the media was immediately collected and stored at −20 °C. To measure the amount of insulin secretion, aliquots of the samples were assayed using an insulin antiserum immunoassay kit (Mercodia, Uppsala, Sweden) according to the manufacturer’s instructions.

### 4.5. Measurement of Caspase-3 Activity

RIN-m5F cells were seeded and cultured at 37 °C, and then treated with BPA or MBP for 24 h. At the end of the treatment (24 h), the cell lysates were incubated at 37 °C with 10 μM Ac-DEVD-AMC, a caspase-3/CPP32 substrate (Promega Corporation, Madison, WI, USA), for 1 h. The fluorescence of the cleaved substrate was measured using a spectrofluorometer (Gemini XPS Microplate Reader, Molecular Devices, San Jose, CA, USA) at an excitation wavelength of 380 nm and an emission wavelength of 460 nm.

### 4.6. Western Blot Analysis

RIN-m5F cells were seeded at 1 × 10^6^ cells/well in a 6-well culture plate and treated with BPA or MBP. At the end of various treatment durations, the levels of protein expression were analyzed by means of Western blot analysis, as previously described [20,21]. In brief, equal amounts of proteins (50 μg per lane) were subjected to electrophoresis on 10% (*w*/*v*) SDS-polyacrylamide gels and transferred onto polyvinylidene difluoride (PVDF) membranes. The membranes were blocked for 1 h in PBST (PBS with 0.05% Tween-20) containing 5% nonfat dry milk. After blocking, the membranes were incubated with the specific antibodies against caspase-3, caspase-7, caspase-9, PARP, p-JNK, p-AMPKα, JNK, AMPKα, GPR 78, GRP 94, CHOP, caspase-12, and β-actin in 0.1% PBST (1:1000) for 1 h. After three additional washes in 0.1% PBST (15 min each), the respective HRP-conjugated secondary antibodies were applied (in 0.1% PBST (1:2500)) for 1 h. The antibody-reactive bands were detected using enhanced chemiluminescence reagents (Pierce™, Thermo Fisher Scientific Inc., Waltham, MA, USA) and analyzed using a luminescent image analyzer (ImageQuant™ LAS-4000; GE Healthcare Bio-Sciences, Uppsala, Sweden). The bands were analyzed via densitometric analysis using ImageJ software and signals normalized to that of the housekeeping control.

### 4.7. Statistical Analysis

Data are presented as the mean ± standard deviation (S.D.) of at least four independent experiments. All data analyses were performed using SPSS software version 12.0 (SPSS, Inc., Chicago, IL, USA). For each experimental condition, the significant difference compared to that of the respective controls was assessed by means of one-way analysis of variance (ANOVA), and Tukey’s post hoc test was performed to identify group differences. A *p*-value of less than 0.05 was considered a significant difference.

## 5. Conclusions

Collectively, our findings demonstrate for the first time that MBP is capable of inducing β-cell cytotoxicity and death via the interaction between JNK and AMPK signals, which regulate ER stress-triggered apoptosis. These observations also provide beneficial evidence suggesting that MBP may be a risk factor for the development of DM.

## Figures and Tables

**Figure 1 ijms-22-04379-f001:**
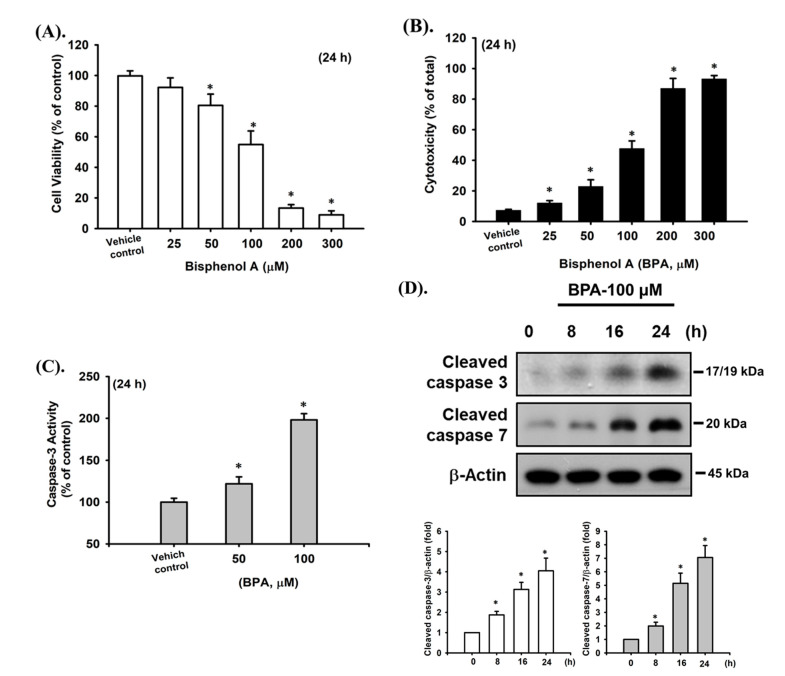
BPA-induced cytotoxicity and apoptosis in RIN-m5F cells. Cells were treated with BPA (0–300 μM) for 24 h. (**A**) Cell viability was detected using the MTT assay. (**B**) Cytotoxicity was determined using the LDH release assay. (**C**) Caspase-3 activity was determined using the caspase-3 activity fluorometric assay kit. (**D**) The protein expression of cleaved caspase-3 and -7 was examined using Western blot analysis, and the quantification was performed using densitometric analysis. Data are presented as the mean ± S.D. of four independent experiments assayed in triplicate. * *p* < 0.05 compared to the vehicle control.

**Figure 2 ijms-22-04379-f002:**
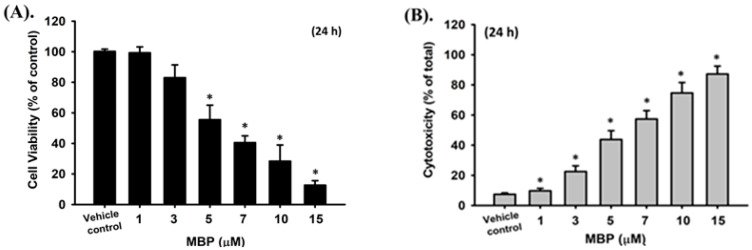
Effects of MBP on cytotoxicity, apoptosis, and insulin secretion in RIN-m5F cells. The cells were treated with MBP (0–15 μM) for 24 h; subsequently. (**A**) Cell viability was detected using the MTT assay. (**B**) Cytotoxicity was determined using the LDH release assay. (**C**) Caspase-3 activity was determined using the caspase-3 activity fluorometric assay kit. (**D**) The protein expression of cleaved caspase-3, -7, and -9 and cleaved PARP proteins was examined using Western blot analysis. The quantification was performed using densitometric analysis. (**E**) In addition, RIN-m5F cells were treated with 5 μM MBP for different time intervals (2–24 h). Cell viability was determined using the MTT assay. (**F**) RIN-m5F cells were treated with MBP (1–5 μM) for 4 h. The insulin secretion stimulated by 2.8 mM or 20 mM-glucose was detected using an immunoassay kit. Data are presented as the mean ± S.D. of four independent experiments assayed in triplicate. * *p* < 0.05 compared to the vehicle control.

**Figure 3 ijms-22-04379-f003:**
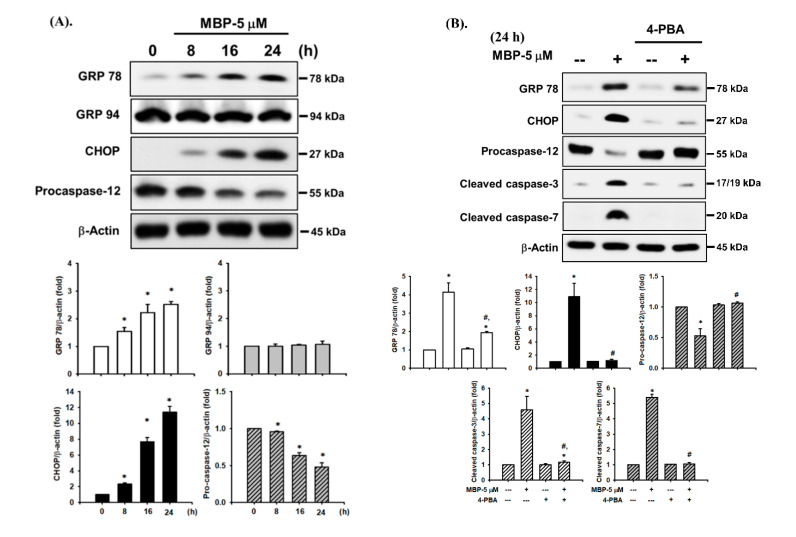
Involvement of ER stress-regulated signaling pathways in MBP-induced β-cell apoptosis. (**A**) RIN-m5F cells were treated with or without MBP (5 μM) for various time intervals (8–24 h). The protein expression of the ER stress-related proteins GRP 78, GRP 94, CHOP, and caspase-12 was examined using Western blot analysis. Additionally, RIN-m5F cells were treated with or without MBP (5 μM) for 24 h in the absence or presence of 3 mM 4-phenylbutyric acid (4-PBA; the pharmacological inhibitor of ER stress, 1 h treatment prior to MBP exposure). (**B**) The protein expression of GRP 78, CHOP, pro-caspase-12, and cleaved caspase-3 and -7 was examined using Western blot analysis. The quantification was performed using densitometric analysis. (**C**) Caspase-3 activity was determined using the caspase-3 activity fluorometric assay kit. (**D**) Cell viability was detected using the MTT assay. Data are presented as the mean ± S.D. of four independent experiments assayed in triplicate. * *p* < 0.05 compared to the vehicle control. # *p* < 0.05 compared to treatment with MBP alone.

**Figure 4 ijms-22-04379-f004:**
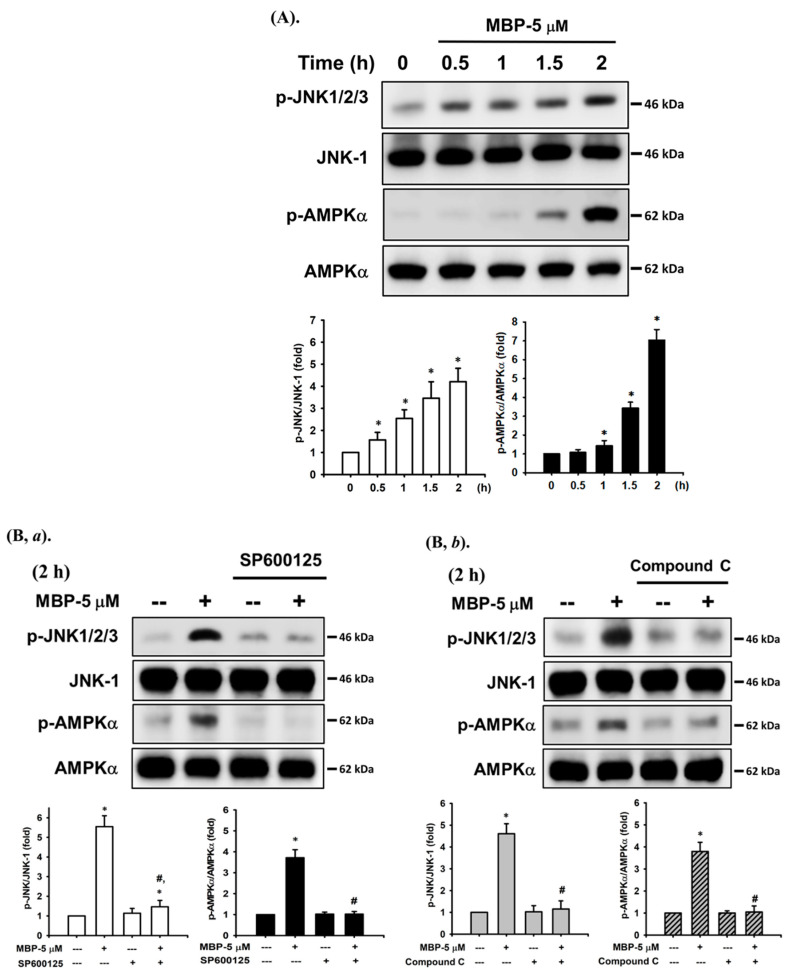
MBP caused the activation of JNK and AMPK signals in β-cells. (**A**) RIN-m5F cells were treated with or without MBP (5 μM) for various time intervals (0.5–2 h). (**B**) RIN-m5F cells were treated with MBP (5 μM) for 2 h in the presence or absence of the specific inhibitor of JNK1/2 (SP600125—10 μM, (**B,*a***)) and AMPKα (compound C—10 μM, (**B,*b***)). The protein expression of phosphorylated JNK1/2 and phosphorylated AMPKα was examined using Western blot analysis. The quantification was performed using densitometric analysis. Data are presented as the mean ± S.D. of four independent experiments assayed in triplicate. * *p* < 0.05 compared to vehicle control.

**Figure 5 ijms-22-04379-f005:**
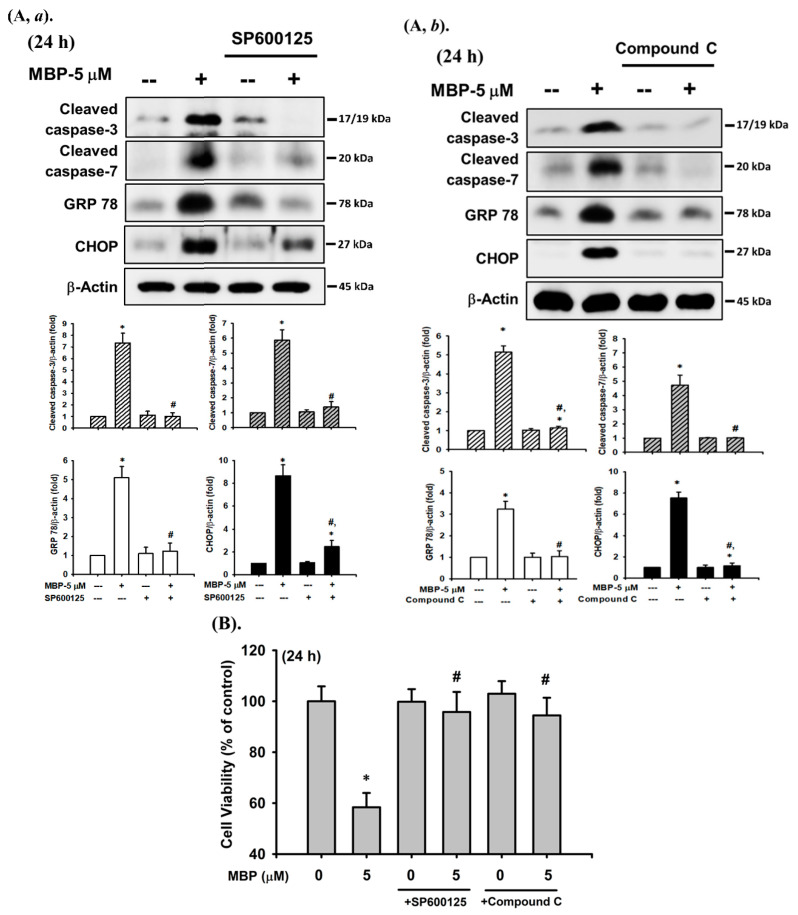
JNK- and AMPKα-mediated signaling pathways played critical roles in MBP-induced apoptosis in β-cells. RIN-m5F cells were treated with MBP (5 μM) for 2–24 h in the presence or absence of the specific inhibitor of JNK1/2 (SP600125—10 μM; (**A,*a***) and **B**) and AMPKα (compound C—10 μM; (**A,*b***) and **B**). (**A**) The protein expression of cleaved caspase-3 and -7, cleaved PARP, GRP 78, and CHOP was determined using Western blot analysis. The quantification was performed using densitometric analysis. (**B**) The cell viability was detected using the MTT assay. Data are presented as the mean ± S.D. of four independent experiments assayed in triplicate. * *p* < 0.05 compared to the vehicle control. # *p* < 0.05 compared to treatment with MBP alone.

## Data Availability

Please contact the corresponding author for reasonable data request.

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
