# Peer review of "4-Methyl-2,4-bis(4-hydroxyphenyl)pent-1-ene, a Major Active Metabolite of Bisphenol A, Triggers Pancreatic β-Cell Death via a JNK/AMPKα Activation-Regulated Endoplasmic Reticulum Stress-Mediated Apoptotic Pathway"

_ijms, 2021, doi:10.3390/ijms22094379_

Round 1
Reviewer 1 Report
The current study by Huang and colleagues investigate the action of MBP, a metabolite of Bisphenol A, on pancreatic beta-cell function.
BPA as a dietary contaminant is reported to generate MBP through the liver, and "might accumulate in mammals upon BPA exposure".
The authors show that both BPA and MBP have significant effects on pancreatic beta cell viability, triggering apoptotic cascades leading to cell death.
Moreover, MBP appears to affect glucose-stimulated insulin secretion in the RIN-m5F beta-cell line. These findings are generally interesting to the beta-cell field, suggesting underlying cytotoxic environmental contaminants may be a contributing factor to the development of diabetes mellitus.
However, an improved introductory section would help clarify and increase the study's relevance and significance.
For example, the authors note that while BPA has been linked to beta-cell dysfunction, MBP has not been investigated in a beta-cell function context.
However, they make no comment as to what concentrations of BPA (presumably via dietary intake) may result in MBP accumulation. Moreover, what concentrations of either of these compounds become physiologically relevant to in vivo pathology, and how do the concentrations used in the current study reflect this?
Additionally, what other metabolites are known to be generated from BPA? Have MBP levels been measured from humans? Is there a link between increased estrogenic activity and plasma insulin?
Specific comments:
All data should be presented as individual points on bar graph
In Figure 1A: what is vehicle control? Is BPA dissolved in a solvent consistent in the control wells?
In Figure 1 Legend, BPA dose should be 0 - 300uM
Please give information in the text as to how and why was 24h timepoint selected
In Figure 2; again what was vehicle control?
Figure 2E notes no change in cell viability at 4 hours in MBP-treated cells, though no 4h timepoint is measured for caspase activity. How was cell viability measured? Is insulin content the same in these cells?
For consistency, insulin secretion should also be measured in remaining cells after 24h MBP treatment.
Insulin secretion should be normalised to insulin content or total protein levels in this case.
Should be IC50, not EC50 if refering to cytotoxicity of MBP.
In Figure 3, for pre-treatment with 4-PBA, was 4-PBA removed at MBP treatment?
How does 4-PBA affect cell viability alone and with MBP?
In Figure 4. Pretreatment for 1h with inhibitors, how were these concentrations selected?
The order of Figure 5 is confusing. Should have 5C follow Figure 4 as these conditions are consistent (2h MBP).
5A and B are presumably 24h MBP treatment as cell viability is decreased, though this is not specifically stated and requires confirmation.
Authors should show cell viability data for 2h and4 h MBP.
Lines:
215, "animal study" should be "studies"
228, "exhibited" not "exhibitd"
233, "its" not "its'"
233, "mammalian" should be "mammals" or "mammalian cells"
235, should be IC50 not EC50
241, "unfold" should be "unfolded"
"Additionally, MBP has been reported to exhibit greater cytotoxicity than BPA." This sentence is not substantiated. No direct comparison was made between these two compounds. Additionally, there is no evidence of correlative concentrations between BPA and MBP.
Author Response
Q1. However, an improved introductory section would help clarify and increase the study's relevance and significance. For example, the authors note that while BPA has been linked to beta-cell dysfunction, MBP has not been investigated in a beta-cell function context. However, they make no comment as to what concentrations of BPA (presumably via dietary intake) may result in MBP accumulation. Moreover, what concentrations of either of these compounds become physiologically relevant to in vivo pathology, and how do the concentrations used in the current study reflect this? Additionally, what other metabolites are known to be generated from BPA? Have MBP levels been measured from humans? Is there a link between increased estrogenic activity and plasma insulin?
Ans.: We have been reorganized and rewritten the first paragraph of ‘3. Discussion’ section to clarify and increase the relevance and significance about MBP in this study, as following:
‘‘BPA is one of the highest production volume chemicals in the world and an environmental risk factor for the development of DM; humans are continually exposed to it via contaminated environments, plastic, and other products [29-31]. BPA has been detected at the concentration of 21-17200 μg/L (approximately 0.1-75.3 μM) in polluted aquatic environments such as river water and landfill leachate [32]. In resin-based composites and sealants in dentistry, BPA levels were reported to range from 0.5 to 84.4 μg per 100 mg of commercial product, and the saliva samples obtained after treatment using these products contained 3.3 to 30.0 μg/mL (approximately 14.5 to 131.4 μM) [33]. In some animal studies, long-term exposure to 0.1-10 mg/kg/day BPA decreased the plasma insulin levels and increased the number of active caspase-3- positive cells in the pancreatic islets (approximately 0.44-43.8 μM/day), which accelerated DM development [34-35]. More importantly, MBP as an active metabolite of BPA, is formed upon coincubation of BPA and liver microsomal and cytosolic fractions (S9 fraction from mammalian liver samples, including human samples) [11]. Consequently, it has been considered that BPA released into the aquatic environment, that gains access in the body through various routes, could be converted to MBP in mammalian liver, triggering a considerably stronger toxicological effect than its parent compound BPA [36-38]. It has been shown that MBP is a more potent binder of the estrogen receptor than BPA by luciferase reporter assays in vitro [11, 13]. Hirao-Suzuki and colleagues have also reported repeated exposure to MBP, but not to BPA, aggressively stimulated the abnormal proliferation in breast cancer cells through the activation of estrogen receptor b-dependent signaling [18]. More importantly, Ishibashi et al. [12] have shown that the 96 h median lethal concentration of MBP in medaka (Oryzias latipes) was estimated to be 1640 and 13900 μg/L (approximately 6.1 and 60.9 μM), respectively. A study by Okuda et al. [13] in an ovariectomized (OVX) female rat model indicated that MBP (1000 μg/kg/day (approximately 3.7 μM/day)) completely reversed the changes caused by OVX, as equivalent to that activity of 17b-estradiol 0.5 μg/kg/day, suggesting that MBP exhibited at least 500-fold higher estrogenic activity than BPA. Furthermore, exposure to MBP 25 μg/L (approximately 0.1 μM) has been found to impair cardiovascular function and induce the development of vascular-cardiovascular disease states in zebrafish [37]. Huang et al. [38] and Liu et al. [19] have also demonstrated that MBP exposure (5-15 μM) induced dysfunction and apoptosis in pulmonary alveolar epithelial cell and neuronal cell. However, the cytotoxic effect and its action mechanism of MBP on b-cells are fully unclear, especially with respect to its possible exposure concentration in mammalians. This study, for the first time, demonstrated that treatment of RIN-m5F cells with MBP significantly induced cytotoxicity in a concentration-dependent manner (ranging from 3 to 15 μM; IC50 was approximately 5 μM), which accompanied with insulin secretion dysfunction and apoptotic events. Moreover, the signaling mechanisms of both JNK and AMPK activation in which contribute to triggering ER stress-mediated apoptosis are involved in MBP-induced b-cell death.’’ (in P9, L226-261 of the revised manuscript).
Q2. All data should be presented as individual points on bar graph
Ans.: We have been according to reviewer’s suggestion to present as individual points on bar graph of all data.
Q3. In Figure 1A: what is vehicle control? In Figure 2; again what was vehicle control?
Ans.: The point of ‘0’ mean ‘Vehicle control’. We have also corrected ‘0’ to ‘Vehicle control’ in Figure 1 and 2 of the revised manuscript.
Q4. Is BPA dissolved in a solvent consistent in the control wells?
Ans.:
- The higher concentration BPA and MBP (the stock solution) used in this study are dissolved in dimethyl sulfoxide (DMSO).
- The control wells were added the maximum volume DMSO of used, which was not more than 1% per well (< 1%).
- In this study, the maximum volume of DMSO added was approximately 5 μL in 1 mL culture medium, and that was not affected cell viability for 24 h exposure (1 μL-DMSO: 101.5 ± 4.3 % of normal control; 3 μL-DMSO: 99.2 ± 3.7 % of normal control; 5 μL-DMSO, 102.3 ± 4.1 % of normal control).
Q5. In Figure 1 Legend, BPA dose should be 0 - 300 μM
Ans.: This mistake has been corrected. Please see P3, L98-99.
Q6. Please give information in the text as to how and why was 24h timepoint selected
Ans.: We have been added the sections in the text to explain ‘how and why was 24h time point selected’, as follows:
- As shown in Fig. 2D, Western blot analysis showed an increase in the expression levels of cleaved forms of caspase-3, -7, and -9 and PARP proteins after 8 h of treatment of RIN-m5F cells with MBP (5 μM) and significantly continued increase to 24 h (in a time-dependent manner). Furthermore, the exposure of RIN-m5F cells to 5 μM MBP for various time intervals (2-24 h) also resulted in a significant reduction the cell viability at 8 h and dramatically gradual reduce to 24 h (2 h, 101.6 ±2 % of control; 4 h, 100.9 ± 3.9% of control; 8 h, 83.6 ± 4.5 % of control; 16 h, 69.6 ± 4.3 % of control; and 24 h, 51.6 ± 6.2 % of control)(Fig. 2E). (in P4 , L120-126 of the revised manuscript);
- As shown in Fig. 3A, the exposure of RIN-m5F cells to MBP (5 μM) markedly triggered the protein expression of ER stress-related molecules including GRP 78 (but not that of GRP 94) and CHOP, as well as the degradation of full-length (55 kDa) caspase-12 (a downstream ER stress molecule) in a time-dependent manner (the statistical increase after 8 h exposure, which was obviously continued increase to 24 h), as like apoptosis-related molecules. (in P5 , L151-155 of the revised manuscript);
- Thus, we next selected 24 h time point to further confirm the relationship between MBP-induced b-cell apoptosis and the activation of ER stress. The detailed description can be seen in the following:
‘To further confirm the relationship between MBP-induced b-cell apoptosis and the activation of ER stress, RIN-m5F cells were pretreated with a pharmacological inhibitor of ER stress (4-phenylbutyric acid, 4-PBA; 3 mM) for 1 h prior to MBP exposure and subsequently exposed to MBP (5 μM) for 24 h. The results showed that MBP-induced activation of ER stress-related molecules, including upregulation of GRP 78, CHOP, and cleavage caspase-3 and 7 protein expression and degradation of pro-caspase-12, was effectively and significantly prevented by 4-PBA (Fig. 3B). Meanwhile, MBP-induced elevation of caspase-3 activity and reduction of cell viability were also significantly attenuated after pretreatment with 4-PBA (Fig. 3C and 3D). These results imply that MBP can induce ER stress-regulated apoptosis, leading to b-cell death.’(Please see P5 , L156-164 of the revised manuscript).
Q7. Figure 2E notes no change in cell viability at 4 hours in MBP-treated cells, though no 4h timepoint is measured for caspase activity. How was cell viability measured? Is insulin content the same in these cells? For consistency, insulin secretion should also be measured in remaining cells after 24h MBP treatment. Insulin secretion should be normalized to insulin content or total protein levels in this case.
Ans.:
- Before the determination of glucose-stimulated insulin secretion in RIN-m5F cells-treated MBP at 4 h, we measured cell viability by ‘MTT assay’. Please see P4, L and Figure 2F of the revised manuscript.
- We have been according to reviewer’s suggestion that insulin secretion data was be normalized by the using total protein levels (Please see Figure 2F).
Q8. Should be IC50, not EC50 if refering to cytotoxicity of MBP.
Ans.: We have been according to reviewer’s suggestion to ‘EC50’ to ‘the half maximal inhibitory concentration (IC50)’ in P3, L115-116 of the revised manuscript.
Q9. In Figure 3, for pre-treatment with 4-PBA, was 4-PBA removed at MBP treatment? How does 4-PBA affect cell viability alone and with MBP?
Ans.:
- ‘RIN-m5F cells were pretreated with 4-PBA (3 mM) for 1 h’ is indicated ‘1 h treatment with 4-PBA prior to MBP exposure’, and then 4-PBA was not removed at MBP treatment. Please see P5, L151-152 and P6, L182-184 of the revised manuscript.
- We have been according to reviewer’s suggestion to add the result of cell viability in RIN-m5F cell-treated with or without MBP (5 μM) for 24 h in the presence or absence of 3 mM 4-PBA. As shown in Fig. 3D, it was not effected cell viability after treatment of RIN-m5F cells with 4-PBA (3 mM) for 24 h (3 mM 4-PBA, 101.5 ± 4.1 % of control). Meanwhile, MBP-induced reduction of cell viability was effectively reversed after r pretreatment with 4-PBA (5 μM MBP, 54.7 ± 6.5 % of control; 5 μM MBP + 3 mM 4-PBA, 90.5 ± 5.5 % of control, p < 0.05). Please see P5, L161-162 and Figure 3D of the revised manuscript.
Q10. In Figure 5. Pretreatment for 1h with inhibitors, how were these concentrations selected?
Ans.:
- The concentration selection of SP600125 (10 μM) and Compound C (10 μM) is according to the study of Yang et al. (2016), which could effectively prevented the molybdenum (Mo)-induced JNK and AMPK activation downstream-regulated ER stress-triggered apoptosis in pancreatic b-cells, and not cause cytotoxicity by itself.
- Thus, we chose these concentrations of SP600125 (10 μM) and Compound C (10 μM) used in this study. The results showed that pretreatment of cells with JNK inhibitor (SP600125, 10 μM) and AMPK inhibitor (Compound C, 10 μM) for 1 h prior to MBP exposure significantly inhibited the levels of protein expression of both phosphorylated JNK and phosphorylated AMPKa (Fig. 4B-a). Similarly, pretreatment with a specific AMPK inhibitor (Compound C, 10 μM) markedly attenuated the activation of both AMPKa and JNK following a 2 h treatment with MBP (Fig. 4B-b). Furthermore, the activation levels of caspase-3 and -7, GRP 78, and CHOP protein induced by following a 24 h treatment with MBP were effectively reversed by pretreatment with a specific JNK inhibitor (SP600125, 10 μM) and AMPK inhibitor (Compound C, 10 μM)(Fig. 5A) as well as the inhibition of cell viability (Fig. 5B). Please see the section of ‘2.4. JNK and AMPK signaling played crucial roles in MBP-induced b-cell apoptosis’ (P5, L165-P7, L196) and Figure 4 and 5 of the revised manuscript).
Reference:
Yang, T.Y., Yen, C.C., Lee, K.I., Su, C.C., Yang, C.Y., Wu, C.C., Hsieh, S.S., Ueng, K.C., Huang, C.F. Molybdenum induces pancreatic beta-cell dysfunction and apoptosis via interdependent of JNK and AMPK activation-regulated mitochondria-dependent and ER stress-triggered pathways. Toxicol. Appl. Pharmacol. 2016, 294, 54-64. https://doi.org/10.1016/j.taap.2016.01.013. 30.
Q11. The order of Figure 5 is confusing. Should have 5C follow Figure 4 as these conditions are consistent (2h MBP).
Ans.: The order of Figure 5C has been changed to Figure 4B. We have been reorganized and rewritten the section of ‘2.4. JNK and AMPK signaling played crucial roles in MBP-induced b-cell apoptosis’ in P5, L165-P6, L190 of the revised manuscript. It can be seen in the following:
‘Pretreatment of cells with JNK inhibitor (SP600125, 10 μM) for 1 h prior to MBP exposure significantly inhibited the levels of protein expression of both phosphorylated JNK and phosphorylated AMPKa (Fig. 4B-a). Similarly, pretreatment with a specific AMPK inhibitor (Compound C, 10 μM) markedly attenuated the activation of both AMPKa and JNK following a 2 h treatment with MBP (Fig. 4B-b).’
Q12. 5A and B are presumably 24h MBP treatment as cell viability is decreased, though this is not specifically stated and requires confirmation.
Ans.: We have been confirmed that Figure 5A and 5B were treated with MBP for 24 h and added the label of ’24 h’ in Figure 5B.
Q13. Authors should show cell viability data for 2h and 4 h MBP.
Ans.: We have been according to reviewer’s suggestion to perform the cell viability assay following 5 μM MBP for different time intervals (2-24 h) in Figure 2E of the revised manuscript. The result can be seen in the following:
‘Furthermore, the exposure of RIN-m5F cells to 5 μM MBP for various time intervals (2-24 h) also resulted in a significant reduction the cell viability at 8 h and dramatically gradual reduce to 24 h (2 h, 101.6 ± 3.2 % of control; 4 h, 100.9 ± 3.9% of control; 8 h, 83.6 ± 4.5 % of control; 16 h, 69.6 ± 4.3 % of control; and 24 h, 51.6 ± 6.2 % of control;)(Fig. 2E).’ in P4, L123-126 and Figure 2E of the revised manuscript.
Q14. Lines:
215, "animal study" should be "studies"
228, "exhibited" not "exhibitd"
233, "its" not "its'"
233, "mammalian" should be "mammals" or "mammalian cells"
235, should be IC50 not EC50
241, "unfold" should be "unfolded"
Ans.: These mistakes have been corrected. We have been carefully corrected and proofreading our revised manuscript with language usage, grammar and syntax by a native English speaker. If further editorial corrections are needed, we will be pleased to revise according to your suggestions.
Q15. "Additionally, MBP has been reported to exhibit greater cytotoxicity than BPA." This sentence is not substantiated. No direct comparison was made between these two compounds. Additionally, there is no evidence of correlative concentrations between BPA and MBP.
Ans.: We have been deleted this sentence to avoid misunderstanding.

Reviewer 2 Report
The authors have prepared a manuscript exploring the cellular effects of a metabolite of bisphenol A (BPA) called MBP on pancreatic beta cells. The authors have performed an experimental study on RIN-m5F cells and have tested cytotoxicity, induction of apoptosis of BPA and MBP. Furthermore, induced ER stress was assessed and the JNK and AMPK signaling pathways were also evaluated in terms of their involvement in apoptosis.
The introduction presents background data and provides key information placing the article in the general context. In three paragraphs, the authors cover the burden of diabetes mellitus, the impact of BPA and MBP and the few studies that have explored the connection between exposure to these substances and apoptosis or carcinogenesis.
The study is methodologically sound, with proper presentation of the materials and methods involved in the experimental work.
The results are concise, clear, and supported by figures that, while repeating some of the data presented in the text, provide a better understanding of the statistical relevance and offer a clear image of the study outcomes.
The discussions are well elaborated and cover several topics. The signaling pathways of apoptosis induction in beta cells are presented in short. Some more background on BPA is given, which might have been better suited in the introduction, however, it directly relates to content explained in the Discussion chapter since it relates to doses/exposure. ER stress and the cellular mechanisms implicated are presented in a large paragraph, while discussing the relevance of MBP exposure as it is referenced in the international literature. The role of JNK and AMPK activation is presented in the end, as well as the innovative characteristic of the results.
The conclusions are very brief and present some of the findings.
Overall, the article is very well written, and I have very few comments:
- Perhaps the main results of the study should be summarized in a short paragraph at the end of Discussions. Alternatively, they could be presented in the Conclusion chapter, if they can be fitted within 5-10 lines.
- Institutional Review Board Statement, Informed Consent Statement, Data Availability Statement are missing from the back matter.
- Some minor grammar/spelling issues should be addressed.
- Some references (website) are not compliant with instructions for authors.
Respectfully submitted,
Author Response
Q1. Perhaps the main results of the study should be summarized in a short paragraph at the end of Discussions. Alternatively, they could be presented in the Conclusion chapter, if they can be fitted within 5-10 lines.
Ans.: We have been according to reviewer’s suggestion to summarize the main results of the study in the ‘5. Conclusion’ section, as following:
‘5. Conclusion
Collectively, our findings elucidate for the first time that MBP is capable of inducing b-cell cytotoxicity and death via the interaction between JNK and AMPK signals, which regulate ER stress-triggered apoptosis. These observations also provide beneficial evidence suggesting that MBP may be a risk factor for the development of DM.’ (in P12, L387-L391 of the revised manuscript).
Q2. Institutional Review Board Statement, Informed Consent Statement, Data Availability Statement are missing from the back matter.
Ans.: These information have been added.
Q3. Some minor grammar/spelling issues should be addressed.
Ans.: We have been carefully corrected and proofreading our revised manuscript with language usage, grammar and syntax by a native English speaker. If further editorial corrections are needed, we will be pleased to revise according to your suggestions.
Q4. Some references (website) are not compliant with instructions for authors.
Ans.: We have been carefully rechecked and corrected the part of ‘References (including website)’ section in the revised manuscript.
